# Antibacterial and Anti-Biofilm Activity of Omega-3 Polyunsaturated Fatty Acids against Periprosthetic Joint Infections-Isolated Multi-Drug Resistant Strains

**DOI:** 10.3390/biomedicines9040334

**Published:** 2021-03-26

**Authors:** Débora C. Coraça-Huber, Stephan Steixner, Alexander Wurm, Michael Nogler

**Affiliations:** 1Research Laboratory for Biofilms and Implant Associated Infections (BIOFILM LAB), Experimental Orthopedics, University Hospital for Orthopedics and Traumatology, Medical University of Innsbruck, Peter-Mayr-Strasse 4b, Room 204, 6020 Innsbruck, Austria; stephan.steixner@i-med.ac.at (S.S.); michael.nogler@i-med.ac.at (M.N.); 2University Hospital for Orthopedics and Traumatology, Medical University of Innsbruck, Anichstraße 35, 6020 Innsbruck, Austria; Alexander.wurm@i-med.ac.at

**Keywords:** implant-related infections, periprosthetic joint infections, *Staphylococcus aureus*, coagulase-negative *Staphylococci*, biofilm, omega-3 polyunsaturated fatty acids, eicosapentaenoic acid, docosahexaenoic acid

## Abstract

**Background:** Implantable medical devices, such as prosthetics, catheters, and several other devices, have revolutionized medicine, but they increase the infection risk. In previous decades, commercially available antibiotics lost their activity against coagulase-negative *Staphylococci* (CoNS) and several other microorganisms. Docosahexaenoic acid (DHA) and eicosapentaenoic acid (EPA) are the two major omega-3 polyunsaturated fatty acids (ω-3 PUFAs) with antimicrobial properties. **Materials and Methods:** In this study, we tested the EPA and the DHA for its antibacterial and anti-biofilm activity in vitro against *Staphylococcus epidermidis*, *Staphylococcus aureus*, and different CoNS as reference strains and isolated from patients undergoing orthopedic treatment for implant infections. The tests were carried out with the strains in planktonic and biofilm form. Cytotoxicity assay was carried out with EPA and DHA using human gingival fibroblasts HGF-1. **Results:** The highest concentration of EPA and DHA promoted the complete killing of *S. epidermidis* 1457 and *S. aureus* ATCC 25923 in planktonic form. The fatty acids showed low activity against *P. aeruginosa*. EPA and DHA completely killed or significantly reduced the count of planktonic bacteria of the patient isolated strains. When incubated with media enriched with EPA and DHA, the biofilm formation was significantly reduced on *S. epidermidis* 1457 and not present on *S. aureus* ATCC 25923. The reduction or complete killing were also observed with the clinical isolates. The pre-formed biofilms showed reduction of the cell counting after treatment with EPA and DHA. **Conclusion:** In this study, the ω-3 PUFAs EPA and DHA showed antimicrobial and anti-biofilm activity in vitro against *S. aureus*, *S. epidermidis*, and *P. aeruginosa*, as well as against multi-drug resistant *S. aureus* and CoNS strains isolated from patients undergoing periprosthetic joint infections (PJI) treatment. Higher concentrations of the fatty acids showed killing activity on planktonic cells and inhibitory activity of biofilm formation. Although both substances showed antimicrobial activity, EPA showed better results in comparison with DHA. In addition, when applied on human gingival fibroblasts in vitro, EPA and DHA showed a possible protective effect on cells cultured in medium enriched with ethanol. Further studies are required to confirm the antimicrobial activity of EPA and DHA against multi-drug resistant strains and pan-drug resistant strains.

## 1. Introduction

Implantable medical devices, such as prosthetics, catheters, and several other devices, have revolutionized medicine, but they increase the infection risk. Indeed, implant infection is one of the most frequent and severe complications associated with the use of biomaterials [1,2]. More than 500,000 types of medical devices have currently entered the global market. Invasive medical devices, including indwelling and implantable devices, represent just a fraction of these.

Implant infections count as nosocomial infections, being a major cause of death and increased morbidity among hospitalized patients. These infections put great pressure on already overwhelmed medical systems in both developed and undeveloped countries [3]. Over the past decades, the orthopedic community, for example, has become more concerned about the management of implant infections, because the number of patients with this serious complication continues to rise. Those infections are difficult to treat, often requiring surgical implant replacement [4]. The mortality rates for periprosthetic joint infections (PJI) are equivalent to the rates for breast cancer and melanoma [5]. 

The most commonly cultured microorganisms in implant infections are coagulase-negative *Staphylococci* (CoNS; primarily *S. epidermidis*), followed by *S. aureus* and mixed flora [6,7,8]. In previous decades, commercially available antibiotics lost its activity against CoNS and several other microorganisms. Strong and sometimes dramatic increases in the percentage of resistant isolates were noted particularly for penicillin, oxacillin, ciprofloxacin, clindamycin, erythromycin, and gentamicin [9]. One additional mechanism contributing to this phenomenon and negatively affecting the antimicrobial susceptibility of CoNS is the potential of these bacteria to produce biofilms [10]. Biofilm formation also explains why some normal flora organisms traditionally considered “harmless” become pathogenic when they grow on the surface of foreign bodies [11]. 

Omega-3 polyunsaturated fatty acids (ω-3 PUFAs), such as eicosapentaenoic acid (EPA) and docosahexaenoic acid (DHA) families, are widely discussed due to their nutritional and human health benefits. In addition, DHA and EPA are two major ω-3 PUFAs with antimicrobial properties [12]. The antibacterial mechanisms of EPA and DHA seem to target the bacterial cell membrane within and at the membrane, such as disruption of electron transport chain, uncoupling of oxidative phosphorylation, cell lysis, inhibition of enzyme activity, impairment of nutrient uptake, peroxidation, and auto-oxidation [13]. Some detrimental effects are attributed to formation of pores in the membrane due to EPA’s and DHA’s detergent properties and their amphipathic structure, which allow the interaction with the cell membrane. They are also more active on killing Gram-positive than Gram-negative bacteria [13,14]; however, EPA and DHA inhibit the growth of Gram-positive and Gram-negative species [15,16,17]. EPA and DHA have an unstable characteristic tending to bind to non-specific proteins, and this non-specific mode of action is also desirable, which reduces the antibiotic resistance [13]. In addition, omega-3 fatty acids have proven anti-inflammatory and antioxidant activity, which can benefit the patients undergoing treatment of implant infections. 

In this study, we tested the eicosapentaenoic acid (EPA) and the docosahexaenoic acid (DHA) for its antibacterial and anti-biofilm activity in vitro against *Staphylococcus epidermidis*, *Staphylococcus aureus*, and different coagulase-negative *Staphylococci* as reference strains and isolated strains obtained from patients undergoing orthopedic treatment for implant infections. Cytotoxicity assay was carried out with EPA and DHA using human gingival fibroblasts HGF-1.

## 2. Materials and Methods

### 2.1. Omega-3 Polyunsaturated Fatty Acids (ω-3 PUFAs) and Control Substance

Docosahexaenoic acid (DHA) and eicosapentaenoic acid (EPA) were tested separately. DHA and EPA (Cayman Chemical Company, Ann Arbor, MI, USA) were diluted in phosphate buffered saline (PBS) pH 7.4 (Carl Roth GmbH, Karlsruhe, Germany) for the obtainment of different concentrations (5 mg/mL, 2.5 mg/L, 1.25 mg/L, 0.625 mg/L, and 0.312 mg/L). As the EPA and DHA were delivered from the manufacturer as a solution diluted in ethanol, we decided to use ethanol mixed with nutrient media as control to ensure that an antibacterial or anti-biofilm activity was related to the fatty acids and not to the ethanol (20 µL ethanol ≥ 99.9% + 80 µL Müller-Hinton broth).

### 2.2. Strains

For this study, we used Staphylococcus epidermidis 1457, Pseudomonas aeruginosa ATCC27853, and Staphylococcus aureus ATCC25923 as reference strains and Staphylococcus haemoliticus, Staphylococcus aureus, Staphylococcus simulans, Staphylococcus lugdunensis, and Staphylococcus warneri as coagulase-negative Staphylococci (CoNS) patient isolated strains. The clinical isolates were obtained from patients undergoing PJI treatment at the Department of Orthopedics and Traumatology of the Medical University Innsbruck, Austria. The protocol used in this study was evaluated and approved by the Human Ethic Committee of the Medical University Innsbruck (AN2017-0072 371/4.24 396/5.11-4361A). The identification of all clinical isolates was carried out using conventional microbiological cultures followed by confirmation of bacterial identification using Matrix Assisted Laser Desorption Ionization-Time of Flight Mass Spectrometry (MALDI-TOF-MS). The Division of Hygiene and Medical Microbiology Department of the Medical University Innsbruck was responsible for the identification of bacterial strains using MALDI-TOF technology under certification ISO EN 9001-2008. After identification, each strain was cryopreserved at −80 °C in special medium until the realization of the tests.

### 2.3. Antibiotic Sensitivity Tests

For the obtainment of antibiotic susceptibility rates, the reference and patient isolated strains were suspended in saline solution (0.85% NaCl *w*/*v* in water) at a McFarland of 0.5 (1–2 × 10^8^ CFU/mL). Using a sterile cotton swab, the solution was inoculated on different agar mediums according to the strain species: *Staphylococci* were inoculated on Mueller–Hinton agar; *Pseudomonas* on Mueller–Hinton agar enriched with 5% horse blood and 20 mg/L ß-NAD (β-nicotinamid adenin dinucleotide) (bioMérieux Austria GmbH, Vienna, Austria). After 15 min of the inoculation, the antibiotic discs (BBL™ Sensi-Disc™ Susceptibility Test Discs, BD Life Sciences, Heidelberg, Germany) were carefully placed on the agar plates. Standardly, 31 different antibiotics were tested (Appendix A). After placing the antibiotic discs, the plates were incubated at 37 °C for 16–20 h. After incubation, the zones of inhibition were measured and the millimeters converted into category of susceptibility (susceptible, intermediate, and resistant) according to EUCAST (European Committee on Antimicrobial Susceptibility Testing) [18]. The Division of Hygiene and Medical Microbiology Department of the Medical University Innsbruck carried out the antibiotic susceptibility tests under certification ISO EN 9001-2008.

### 2.4. Activity of ω-3 PUFAs on Planktonic Cells

For the antimicrobial activity tests using EPA and DHA against bacteria in planktonic form, 5 mL of fresh media containing each microorganism were incubated overnight at 37 °C for activation of the strains. After overnight growth, the solution containing the strains was washed under centrifugation 2× each at 2000× *g* with 5 mL fresh phosphate buffer solution (PBS). Then, 20 µL of each different concentration of EPA and DHA (5, 2.5, 1.25, 0.625, and 0.312 mg/mL) were added into wells of a 96-well plate filled with fresh 200 µL media containing 5 µL of each bacterial solution (approximately 10^6^ CFU). After the preparation, the plates were incubated for 24 h at 37 °C. After incubation, we inoculated 10 µL of the content of each well on a fresh Mueller–Hinton agar plate using a spiral platter, and incubated for 37 °C for 24 h. After 24 h, the plates were counted for colony forming units (CFU).

### 2.5. Activity of ω-3 PUFAs for Inhibition of Biofilm Formation-Biofilm Formation in Medium Enriched with EPA and DHA

Each strain was incubated in Mueller–Hinton broth for 24 h at 37 °C overnight. After incubation, each inoculum was diluted to 10^6^ CFU/mL, and 200 uL of the diluted suspension added into wells of a new multi-well plate. The medium was enriched with 20 µL of DHA and EPA at concentrations of 5 mg/L, 1.25 mg/L and 0.312 mg/L. In addition, stainless steel discs (DIN9021, stainless steel A2, size M2, diameter 5.9 mm) were placed individually in each well as a substrate for the biofilm formation. The plates were incubated in an orbital shaker (Edmund Bühler GmbH, Bodelshausen, BW, Germany) for 48 h at 37 °C in a moist chamber for the biofilm formation. After the incubation, the supernatant was removed and the discs washed with fresh PBS for the removal of planktonic bacteria. The discs were added to a tube containing PBS and sonicated (Bactosonic, BANDELIN, Berlin, Germany) for 3 min on the highest level (100%) for the detachment of biofilms. After this step, 10 µL of the sonication fluid were plated on fresh Mueller–Hinton agar plates using a spiral platter. The plates were incubated at 37 °C, and the CFUs counted after 24 h.

### 2.6. Activity of ω-3 PUFAs on Biofilm Killing-Treatment of Biofilms with EPA and DHA

Each strain was grown in Mueller–Hinton broth for 24 h at 37 °C overnight. After incubation, the inoculum was diluted to 10^6^ CFU/mL, and 200 µL of the diluted suspension added into wells of a multi-well plate. In addition, stainless steel discs (DIN9021, stainless steel A2, size M2, diameter 5.9 mm) were individually placed in each well. The plates were incubated in an orbital shaker (Edmund Bühler GmbH, Bodelshausen, BW, Germany) for 48 h at 37 °C in a moist chamber for the biofilm formation. After biofilm formation, the discs were washed in fresh PBS for removal of planktonic cells. Fresh Mueller–Hinton broth (180 µL) with added DHA/EPA (20 µL) in different concentrations (5 mg/L, 1.25 mg/L and 0.312 mg/L) were added to the wells containing the discs, and incubated at 37 °C for 24 h. After the incubation, the supernatant was removed and the discs washed with fresh PBS for the removal of planktonic bacteria. The discs were added to a tube containing fresh PBS and sonicated (Bactosonic, BANDELIN, Berlin, Germany) for 3 min on the highest level (100%) for the detachment of biofilms. After this step, 10 µL of the sonication fluid were inoculated on Mueller–Hinton agar plates using a spiral platter. The plates were incubated at 37 °C, and the CFUs counted after 24 h.

### 2.7. Cytotoxicity Assay of EPA and DHA Using Human Gingival Fibroblasts HGF-1

Human gingival fibroblasts HGF-1 (ATCC^®^ CRL-2014™, Manassas, VI, USA) were used to access the cytotoxic effects of EPA and DHA on eucaryotic cells. For that, the cells were first incubated in a solution of Dulbecco’s Modified Eagle’s Medium (DMEM; Thermo Fisher Scientific, Waltham, MA, USA) with 10% fetal calf serum (FCS; Merck KGaA, Darmstadt, Germany) plus 5.5 mL penicillin–streptomycin (Merck KGaA, Darmstadt, Germany) until the cells reached 90% of confluence at 37 °C, 5%CO_2_, and 95% humidity. After that, the culture medium was removed and 10 mL of phosphate buffered saline (Ca and Mg free; Merck KGaA, Darmstadt, Germany) was added for rinsing and, after gently shaking, completely removed. For the removal of cells, 5 mL of TrypLe™ Express enzyme (1X; Thermo Fisher Scientific, Waltham, MA, USA) with no phenol red was added until the cells were evenly covered. The cells were incubated at 37 °C for 5 min. The detachment of the cells was controlled under inversed light microscopy. Then, 10 mL of DMEM was added and the cells were collected by gently pipetting and the solution transferred to a fresh centrifuge tube. The solution was centrifuged at 200 G for 5 min. The cells were counted using a Neubauer improved chamber, and 10^5^ cells/mL were diluted into fresh DMEM; 100 µL of the cell suspension was added to wells of a sterile 96-well plate. The plate was incubated at 37 °C, 5%CO_2_, and 95% humidity until 90% of confluence was reached. After the incubation, the medium was discarded, and 100 µL of medium added to different concentrations of EPA and DHA (5, 1.25, and 0.3 mg/L, and of medium containing ethanol without EPA and DHA as well as medium without ethanol as control) was added. The plates were again incubated for 2 h at 37 °C, 5%CO_2_, and 95% humidity. After the incubation, the medium was discarded and 100 µL of fresh medium was added in each well including the control wells. Then, 50 µL of tetrazolium salt (XTT; (Merck KGaA, Darmstadt, Germany) solution was added in each well for checking the viability of the cells via cleavage of tetrazolium salts to formazan by the succinate–tetrazolium reductase system, which belongs to the respiratory chain of the mitochondria, and is only active in metabolically intact cells. The plates were then incubated for 5 h, protected from light, at 37 °C, 5%CO_2_, and 95% humidity. After incubation, the plates were gently shaken, and 100 µL of each well was removed and added to a fresh 96-well plate for spectrophotometric measurement using OD455 nm.

### 2.8. Data Analysis

The counting of CFUs from planktonic and bacteria in biofilm form was statistically analyzed using two-way ANOVA with Turkey’s multiple comparisons tests (*p* < 0.0001). The data obtained from the viability tests were analyzed using ordinary one-way ANOVA and Turkey’s multiple comparisons test (*p* < 0.0001). The data are stated as mean ± SEM. For the analysis we used GraphPad Prism 8.0.1 (GraphPad Software, Inc., San Diego, CA, USA). The viability tests were made in duplicate. All the susceptibility tests were repeated twice and carried out in triplicates.

## 3. Results and Discussion

### 3.1. Antibiotic Sensitivity Tests

The antibiotic sensitivity tests showed no drug resistance of the reference strain *S. aureus* ATCC 25923, resistance only to ceftazidime by *S. epidermidis* 1457, and intermediate multi-drug resistance by *P. aeruginosa* ATCC 27853. The patient isolated strains showed multi-drug resistance for at least four different antibiotic substances (Appendix A).

The potential antimicrobial activity of fatty acids is attracting attention due to their potency and broad spectrum of activity and the lack of classical resistance mechanisms against its mechanisms of action [13,19,20]. In particular, various long-chain polyunsaturated fatty acids (LC-PUFAs), which are found naturally at high levels in many marine organisms [21], have been shown to exert highly potent activity against Gram-positive and Gram-negative bacteria, including eicosapentaenoic acid (EPA) and [22,23] docosahexaenoic acid (DHA) [24,25,26,27]. In addition, PUFAs deserve special consideration for oral and topical application, because they are often associated with anti-inflammatory properties, which may provide supplementary benefit during therapy [28].

### 3.2. Activity of ω-3 PUFAs on Planktonic Cells

In accordance with several studies [20,29,30,31,32,33], the anti-bacterial activity in vitro of EPA and DHA against *S. aureus* and coagulase-negative *Staphylococci* confirmed in our results. In this study, all the concentrations of EPA tested were able to completely eliminate *S. aureus* ATCC 25923 when cultivated in planktonic form (Figure 1B). When applied on *S. epidermidis* 1457, only the concentration of 0.312 mg/L of EPA was not able to completely kill the bacteria, but showed a reduction of 1.28 × 10^9^ CFUs compared to the control group (Figure 1A). Using EPA in the highest concentrations of 5 and 2.5 mg/L against *P. aeruginosa* ATCC 27853 a reduction of 2.63 × 10^9^ CFUs were observed in comparison to control (Figure 1C). When treated with DHA, a complete killing was observed on *S. aureus* ATCC 25923 when using the concentrations of 5 and 2.5 mg/L and significant reduction from 1.3 × 10^9^ to 8.7 × 10^8^ of the *S. epidermidis* 1457 when treated with lower concentrations (Figure 1B). DHA was able to eliminate all *S. epidermidis* 1457 in a concentration of 5 mg/L and showed reduction of 1.3 × 10^9^ and 1.29 × 10^9^ CFUs at 2.5 and 1.25 mg/L. The concentrations of 0.625 and 0.312 mg/L were not able to significantly reduce the bacterial count (1 × 10^9^ and 8.7 × 10^8^ reduction to control; Figure 1A). When used against *P. aeruginosa* ATCC 27853, the highest concentration of DHA reduced the CFU in 2.12 × 10^9^. The other concentrations did not show activity against this strain (Figure 1C).

The killing and reduction of CFU was also observed when using EPA and DHA on patient isolates cultured in planktonic form. *S. haemolyticus* was completely eliminated when treated with EPA on the concentrations from 5 to 0.625 mg/L. EPA of 0.312 mg/L was not able to completely kill the strains, but showed a reduction of 1.69 × 10^8^ CFUs compared to the control (Figure 2A). EPA in all concentrations killed all *S. simulans* and *S. lugdunensis* (Figure 2B,C). *S. warneri* was completely eliminated by EPA in the concentrations of 5 to 0.625 mg/L. EPA 0.312 mg/L was able to reduce the bacterial count of *S. warneri* in 7.1 × 10^8^ CFUs compared to the control (Figure 2D). DHA showed more activity against *S. simulans* and *S. lugdunensis*, killing all the strains when used in the concentrations of 5 to 0.625 mg/L. The concentration of 0.312 mg/L, for both strains, was effective in reducing the bacterial count in 7.10 × 10^8^ CFUs (Figure 2B,C). *S. haemolyticus* and *S. warneri* were completely eliminated by DHA when treated with the concentrations of 5 and 2.5 mg/L. The concentrations of 1.25 and 0.625 mg/L showed a reduction of 1.7 × 10^8^ and 1.14 × 10^8^ CFUs for *S. haemoliticus* and 1.60 × 10^9^ and 1.52 × 10^9^ CFUs of *S. warneri*. DHA 0.312 mg/L showed no killing or reduction effect on *S. haemolyticus* and *S. warneri* (Figure 2A,D).

When tested against the strains in planktonic form, almost all concentration of EPA (with exception of the lowest concentrations) were effective and killed all the reference *S. epidermidis* and CoNS patient isolates strains. DHA, on the other hand, showed the same effectivity only when used in higher concentrations. The increased antimicrobial activity followed by the increase in concentration was observed by Le and colleagues (2017) in a study using EPA against *S. aureus* Newman and *Bacillus cereus* NCIMB 9373. With their data, the authors support the development of EPA as a possible new antibacterial agent. The advantages of EPA as an antimicrobial agent are related to its favorable potency against Gram-positive pathogens, showing a lack of rapid selection of bacterial strains with reduced susceptibility or resistance. In this study from Le and colleagues (2017), an interesting fact is the detection of 260 nm absorbing material (A260), which can indicate bacterial membrane perturbation and an increase in membrane permeability, a possible cause of bacterial death when cultured in media rich in n-3 PUFAs fatty acids. The authors detected in their experiments that greater concentrations of EPA led to the detection of greater quantities of A260-absorbing material released from the species tested. Importantly, the increasing quantities of A260-absorbing material coincided with reductions in viable CFU/mL in the suspensions. Taken together, these observations suggest membrane disruption and probable cell lysis of the bacterial cells killed by EPA [30].

We observed less antimicrobial activity of EPA and DHA against Gram-negative *P. aeruginosa* strains, when compared with the Gram-positive strains tested. Similarly, Shin and colleagues (2007) also detected less susceptibility of Gram-negative strains when treated with ω-3 PUFAs. In accordance to the authors, we relate the higher resistance of Gram-negative strains against EPA and DHA to the presence of the outer membrane surrounding the cell wall, which restricts diffusion of hydrophobic compounds through its lipopolysaccharide covering [31,34,35]. Despite the lower activity of EPA and DHA observed by Shin and colleagues, they were able to establish the minimal inhibitory concentration (MIC) of EPA and DHA for *P. aeruginosa* KCTC 2004. Concentrations were 350 mg/mL for EPA and 250 mg/mL for DHA [31]. As in our study, the maximal concentration used was of 5 mg/L; we believe that further tests using higher concentrations of the fatty acids may be important to analyze their activity also against Gram-negative bacteria.

Although in vitro tests are important for the analysis and development of new antimicrobial substances, it is also important to understand that in vitro tests are limited and do not reproduce the complexity of an infected tissue. The surrounding tissues, cellular response to the bacteria, presence of bacterial toxins, and hosts immune responses, amongst other factors, are also relevant. Once in vitro tests may show less activity of omega-3 fatty acids against Gram-negative strains, in vivo tests can show a different outcome. Pierre and colleagues (2007) tested a diet rich in EPA and DHA between other omega-3 fatty acids for the treatment of *P. aeruginosa* lung infections. Their study showed that although the diet did not influence the lung bacterial load, the mice fed with omega-3 diet had a better survival when compared with mice fed with an omega-6 rich diet. The mice fed with omega-3 also showed a significant improvement of the distal alveolar fluid clearance, which was beneficial and facilitated the host immune responses and resolution of the infection [36].

### 3.3. Activity of ω-3 PUFAs for Inhibition of Biofilm Formation—Biofilm Formation in Medium Enriched with EPA and DHA

The biofilm formation in medium enriched with EPA and DHA were carried out using *S. epidermidis* 1457, *S. aureus* ATCC 25923, and the clinical isolates *S. haemolyticus* and *S. aureus*. Medium enriched with EPA in the concentration of 5 mg/L and DHA at 5 mg/L inhibited the proliferation of *S. epidermidis* 1457 bacterial cells in 2logs. For DHA as well as for EPA, the concentrations of 1.25 and 0.312 mg/L were not able to inhibit the biofilm formation by this strain (Figure 3A). *S. aureus* ATCC 25923 was more susceptible to the presence of EPA and DHA in the nutrient media. EPA inhibited completely the biofilm formation of *S. aureus* ATCC 25923 when cultivated under all concentrations tested. The same was observed for *S. aureus* ATCC 25923 strains growing on media enriched with DHA in the concentrations of 5 and 1.25 mg/L. The lower concentration of 0.312 mg/L was not able to inhibit biofilm growth by this strain (Figure 3B).

The biofilm formation by clinical strains were highly influenced by EPA and DHA in high concentration. *S. haemolyticus* biofilm cell counting was of maximal log 10^1^ and log 10^2^ when growing in media enriched with EPA 5 and DHA 5 mg/L. The lower concentrations were not able to inhibit the biofilm formation (Figure 3C). In an opposite way, patient isolated *S. aureus* formed no biofilms when incubated with all concentrations of EPA and with DHA 5 and 1.25 mg/L. The lower concentration of DHA was not able to inhibit the biofilm formation by this strain (Figure 3D).

### 3.4. Activity of ω-3 PUFAs on Biofilm Killing-Treatment of Biofilms with EPA and DHA

To check if EPA and DHA were able to kill already formed biofilms, we incubated 48 h old biofilms in media containing different concentrations of the omega-3 fatty acids. The highest concentration of EPA and DHA (5 mg/L) was able to reduce the biofilm cell counting of *S. epidermidis* 1457 in 2 logs when compared with the untreated group. EPA in the concentrations of 1.25 and 0.312 mg/L were able to reduce the biofilm cell counting in approximately 1 log, while DHA at the same concentrations showed no anti-biofilm activity (Figure 4A). EPA 5 and 1.25 mg/L were able to kill all the biofilms and EPA 0.312 mg/L to reduce the biofilms formed by *S. aureus* ATCC 25923. DHA 5 mg/L killed all the biofilms formed and DHA 1.25 and 0.312 mg/L reduced all the biofilms formed by *S. aureus* ATCC 25923 (Figure 4B).

EPA and DHA were also able to kill or reduce the biofilms formed by clinical isolates. Biofilms formed by *S. haemolyticus* were reduced by 4 logs when treated with EPA 5 mg/L. Differently from the results with planktonic bacteria treated with EPA 5 mg/L where complete kill was observed, some strains showed only reduction on biofilm formation when treated with this concentration. We associate this phenomenon to the fact that some strains produce more slime-rich biofilms, therefore having more protection against antimicrobial substances. While ω-3 PUFAs kill the bacterial cells by altering the fatty acids of the membrane, it may not influence the protein-rich matrix. This is an indication that higher doses of ω-3 PUFAs may be tested for efficient anti-biofilm activity or even the association of ω-3 PUFAs with proteinases to increase the permeability in biofilms. The other concentrations showed no anti-biofilm effect on this strain. The same could be observed when treating the *S. haemolyticus* biofilms with all concentrations of DHA (Figure 5A). Patient isolated *S. aureus* biofilms were strongly reduced by EPA 5 and 1.25 mg/L, killed by DHA 5 mg/L, and reduced by DHA 1.25 mg/L. The lowest concentration of EPA and DHA showed no anti-biofilm effect (Figure 5B). EPA 5 mg/L showed anti-biofilm activity against clinical isolates from *S. simulans*, *S. warneri*, and *S. lugdunensis*, where a reduction of 3–4 logs could be observed on the biofilm cells. A certain reduction could also be observed on the cell counting of the biofilms after treatment with EPA 1.25 mg/L and with DHA 5 and 1.25 mg/L. The lowest concentration of EPA and DHA used showed no anti-biofilm activity (Figure 5C,E).

The anti-biofilm activity of EPA and DHA has been evaluated in different setups: direct killing of in vitro formed biofilms [29,33,37], in vivo using chronic infection models [38], or as coating substances for implants to avoid the biofilm formation [33]. In accordance with our results, all these studies showed the effectiveness of EPA and DHA, as well as of other ω-3 PUFAs, in inhibiting the biofilm formation, or killing an already formed biofilm by different strains. Differently to bacteria in planktonic stages, the bacteria incased in biofilms are less susceptible to antimicrobial substances, including EPA and DHA. Our studies, as well as the results from other authors, suggest that higher concentrations of the ω-3 PUFAs may be used to achieve an efficient anti-biofilm activity.

### 3.5. Cytotoxicity Assay of EPA and DHA Using Human Gingival Fibroblasts HGF-1

We observed a decrease on viability of the cells treated with EPA and DHA 1.25 and 0.312 mg/L. A strong decrease of viability was observed on the cells treated with medium mixed with ethanol. A decrease of viability could be observed on the cells treated with EPA and DHA 5 mg/L (Figure 6A,B). Taking into consideration that all the EPA and DHA solutions contained ethanol originated from its manufacture, we believe that the higher the concentration of EPA and DHA, the less aggressive the ethanol is for the cells, which could suggest a protective effect of the fatty acids on the human fibroblasts. Although more tests may be done to evaluate the biocompatibility of EPA and DHA when applied in high concentrations in vitro using eucaryotic cells, our tests suggest an increased protective effect of EPA and DHA on human gingival fibroblasts. The medium used as control was enriched with ethanol. The reason was that originally, the fatty acids were delivered from the manufacturer in ethanol solution. To be sure that the antimicrobial activity of our concentrations was due to the fatty acids and not an effect of the ethanol, we cultivated not only the bacteria, but also the fibroblasts in medium enriched with ethanol. We detected a change on the morphology and less viability of the fibroblasts in culture with EPA and DHA, but realized that as high as the concentrations of the fatty acids were, so was the viability of the cells (Figure 6A,B). There is a big difference in testing the biocompatibility and cell activity in vitro and in vivo. However, in an interesting study carried out by Wang and colleagues (2017), ω-3 PUFA enriched diet was applied in mice subjected to acute and chronic ethanol feeding plus a single binge as model to induce liver injury with adipose lipolysis. The authors demonstrated that ω-3 PUFA enrichment ameliorated alcoholic liver injury by suppressing ethanol-stimulated adipose lipolysis. In adipocytes, ω-3 PUFA activated membrane G-protein receptor 120 (GPR120) and increased adenosine monophosphate-activated protein kinase (AMPK) activity via Ca2+/CaMKKβ signaling, contributing to ameliorating ethanol-induced adipose dysfunction and liver injury [39].

Taking into consideration the antibiotic susceptibility, we did not recognize any relation between the antimicrobial resistance rates with the susceptibility of the tested strains to EPA and DHA. The fatty acids were effective in killing or reducing the growth of reference and clinical isolated strains in planktonic and biofilm form. The reference and clinical isolated strains tested here were multi-drug resistant (Appendix A). The only strain that showed less susceptibility to EPA and DHA was *P. aeruginosa*. In this case, as already discussed, we associate the less efficacy of the fatty acids to the presence of cell wall and membrane morphology, and not to its antibiotic resistance profile. Although further tests should be carried out with extensively-drug resistant and pan-drug-resistant strains, the antimicrobial activity of EPA and DHA showed no selectivity and was effective against all kind of strains. The use of fatty acids alone or as adjuvant for the treatment of infections could be advantageous in improving the activity of already available antimicrobial substances. As example, Zhou and colleagues (2019) evaluated the synergistic impact of vancomycin and omega-3 fatty acids against osteomyelitis in a *Staphylococcus aureus*-induced rat model of osteomyelitis. In this case, omega-3 fatty acids were orally applied. Combined supplementation with vancomycin and omega-3 fatty acids significantly reduced bacterial growth in bone. In addition, the bone infection levels and histopathological score were reduced. In summary, Zhou and colleagues affirm that the combined treatment with vancomycin and omega-3 fatty acids was effective against bacterial growth and bone infection compared to monotherapy with vancomycin or omega-3 fatty acids [40].

## 4. Conclusions

In this study, the ω-3 PUFAs EPA and DHA showed antimicrobial and anti-biofilm activity in vitro against *S. aureus*, *S. epidermidis,* and *P. aeruginosa* as reference strains, as well as against multi-drug resistant *S. aureus* and CoNS strains isolated from patients undergoing PJI treatment. Higher concentrations of the fatty acids showed killing activity on planktonic cells and inhibitory activity of biofilm formation. Although both substances showed antimicrobial activity, EPA showed better results in comparison with DHA. In addition, when applied on human gingival fibroblasts in vitro, EPA and DHA showed a certain protective effect on cells cultured in medium enriched with ethanol. Further studies are required to confirm the antimicrobial activity of EPA and DHA against multi-drug resistant strains and pan-drug resistant strains. Biocompatibility tests using high concentrations of EPA and DHA are also recommended. Besides the need of further tests, this study is enough to confirm the potential of ω-3 PUFAs omega-3 as antimicrobial and anti-biofilm substances against PJI causative strains. The use of these substances locally for wound healing treatments or directly in the surgical sites during joint replacement surgeries may be an indication and should be further investigated. Furthermore, the combination of local therapy and enriched diet with omega-3 fatty acids could be of advantage in the treatment of implant-related infections.

## Figures and Tables

**Figure 1 biomedicines-09-00334-f001:**
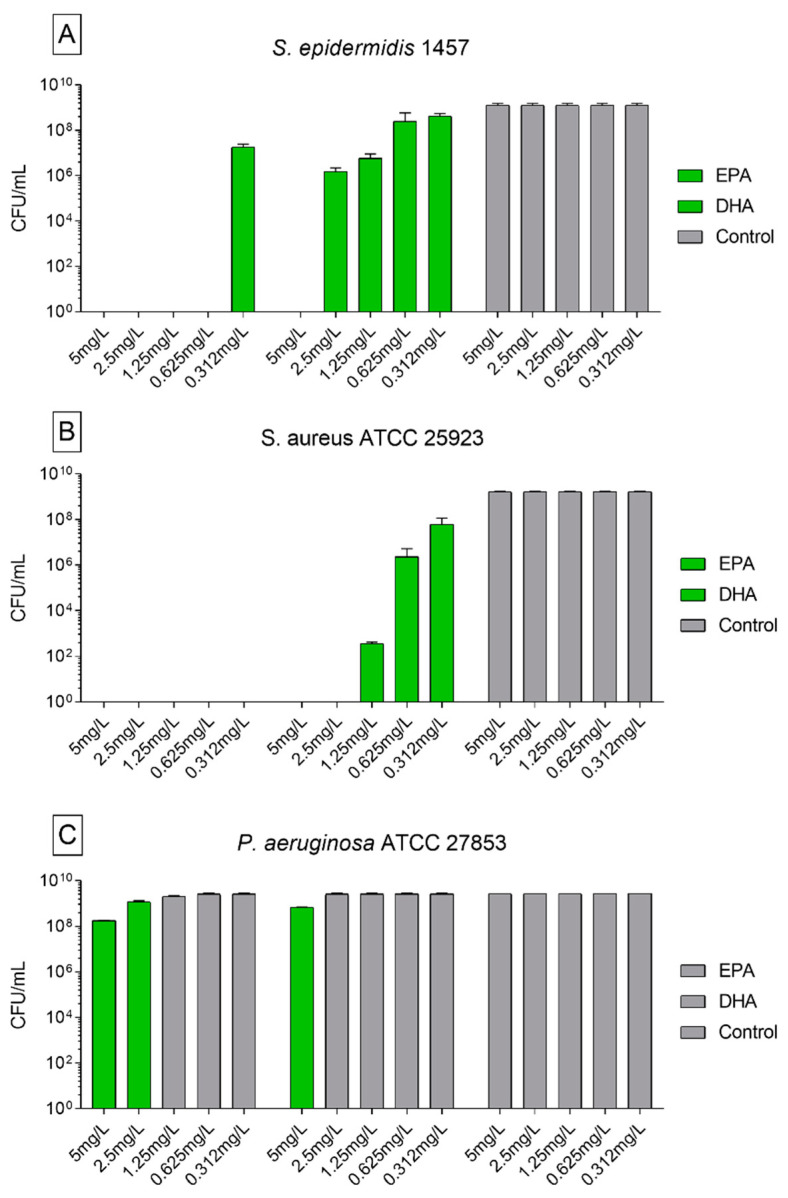
Activity of ω-3 PUFAs on planktonic cells analyzed by counting of colony forming units of reference strains incubated with EPA, DHA, and only with nutrient medium as control. Initial CFU inoculum (approximately 10^6^ CFU). The data are stated as mean ± SEM. (**A**) *S. epidermidis* 1457; (**B**) *S. aureus* ATCC 25923; and (**C**) *P. aeruginosa* ATCC 27853. In green, values significantly different from control group *S. epidermidis* 1457 and *S. aureus* ATCC 25,923 (*p* < 0.0001) and *P. aeruginosa* ATCC 27853 (5 mg/L *p* < 0.001 other concentrations *p* < 0.05). Statistically analyzed with two-way ANOVA and Turkey’s multiple comparison tests. All the tests were repeated twice and carried out in triplicates.

**Figure 2 biomedicines-09-00334-f002:**
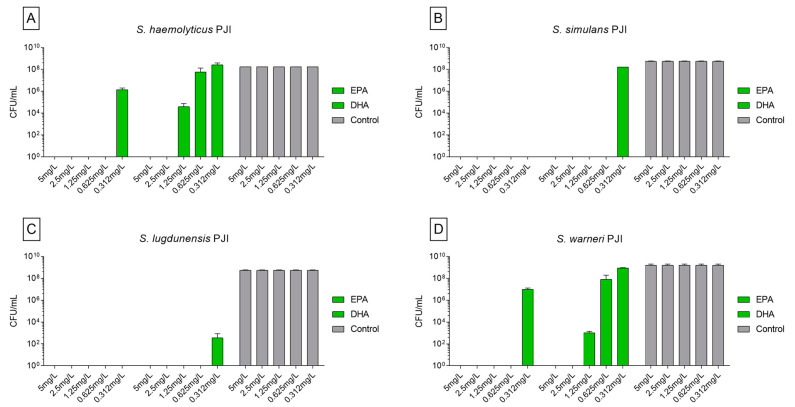
Activity of ω-3 PUFAs on planktonic cells analyzed by counting of colony forming units of patient isolated strains incubated with EPA, DHA, and only with nutrient medium as control. The data are stated as mean ± SEM. (**A**) *S. haemolyticus* PJI patient isolate; (**B**) *S. simulans* PJI patient isolate; (**C**) *S. lugdunensis* PJI patient isolate; (**D**) *S. warneri* PJI patient isolate. In green, values significantly different from control group (*S. haemolyticus* and *S. warneri p* < 0.05; *S. simulans* and *S. lugdunensis p* < 0.0001). Statistical analyzed with two-way ANOVA and Turkey’s multiple comparison tests. All the tests were repeated twice and carried out in triplicates.

**Figure 3 biomedicines-09-00334-f003:**
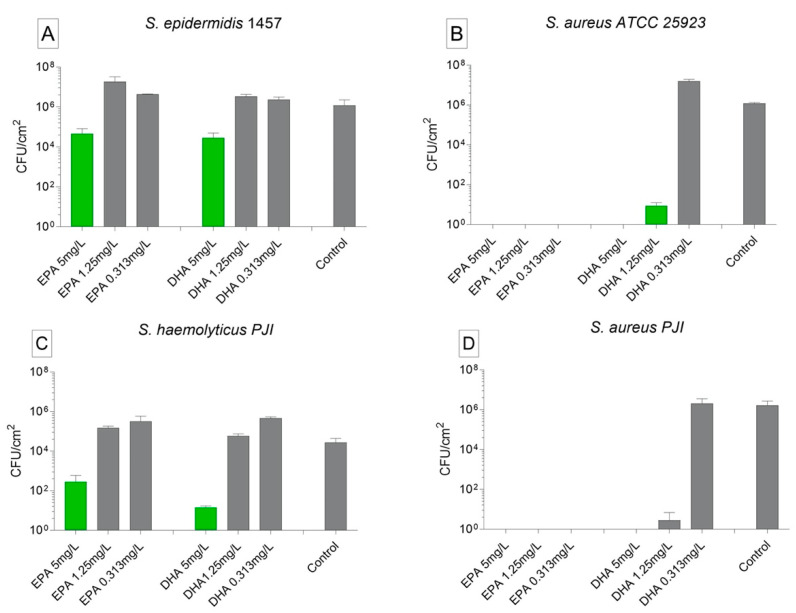
Activity of ω-3 PUFAs for inhibition of biofilm formation—biofilm formation in medium enriched with EPA and DHA, and only with nutrient medium as control. The data are stated as mean ± SEM. (**A**) *S. epidermidis* 1457; (**B**) *S. aureus* ATCC 25923; (**C**) *S. haemolyticus* PJI isolated; (**D**) *S. aureus* PJI isolated. In green, values significantly different from control group (*p* < 0.0001). Statistical analyzed with two-way ANOVA and Turkey’s multiple comparison tests. All the tests were repeated twice and carried out in triplicates.

**Figure 4 biomedicines-09-00334-f004:**
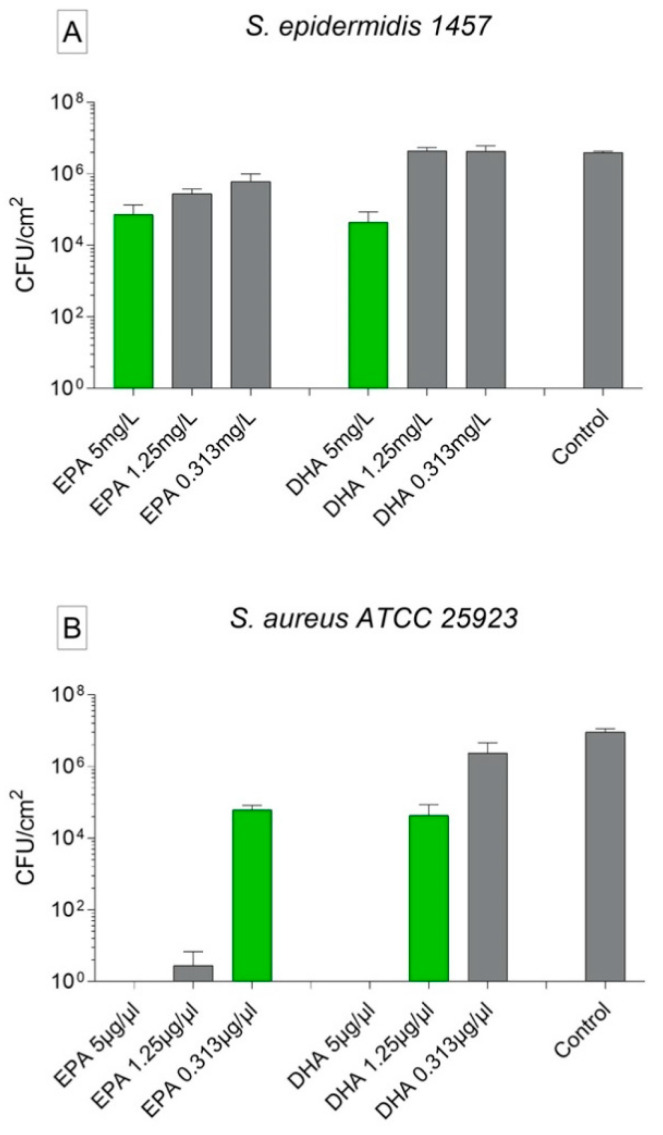
Activity of ω-3 PUFAs on biofilm killing-treatment of biofilms with EPA and DHA, and only with nutrient medium as control. The data are stated as mean ± SEM. (**A**) *S. epidermidis* 1457; (**B**) *S. aureus* ATCC 25923. In green, values significantly different from control group (*p* < 0.0001). Statistically analyzed with two-way ANOVA and Turkey’s multiple comparison tests. All the tests were repeated twice and carried out in triplicates.

**Figure 5 biomedicines-09-00334-f005:**
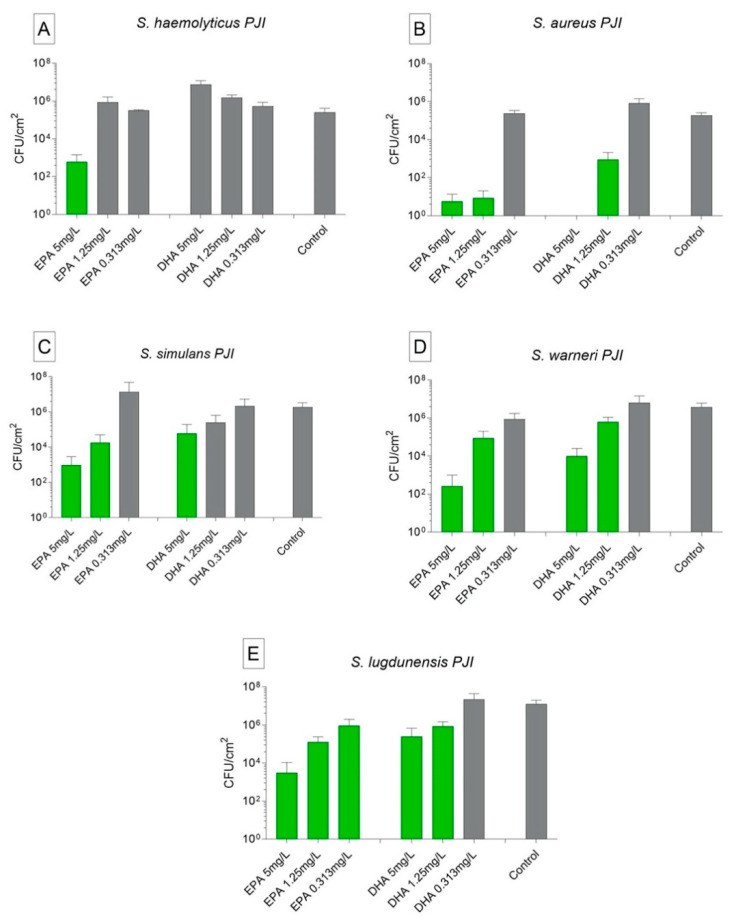
Activity of ω-3 PUFAs on biofilm killing-treatment of biofilms with EPA and DHA, and only with nutrient medium as control. The data are stated as mean ± SEM. (**A**) *S. haemolyticus* PJI isolated; (**B**) *S. aureus* PJI isolated; (**C**) *S.simulans* PJI isolated; (**D**) *S. warneri* PJI isolated; (**E**) *S. lugdunensis* PJI isolated. In green, values significantly different from control group (*p* < 0.0001). Statistically analyzed with two-way ANOVA and Turkey’s multiple comparison tests. All the tests were repeated twice and carried out in triplicates.

**Figure 6 biomedicines-09-00334-f006:**
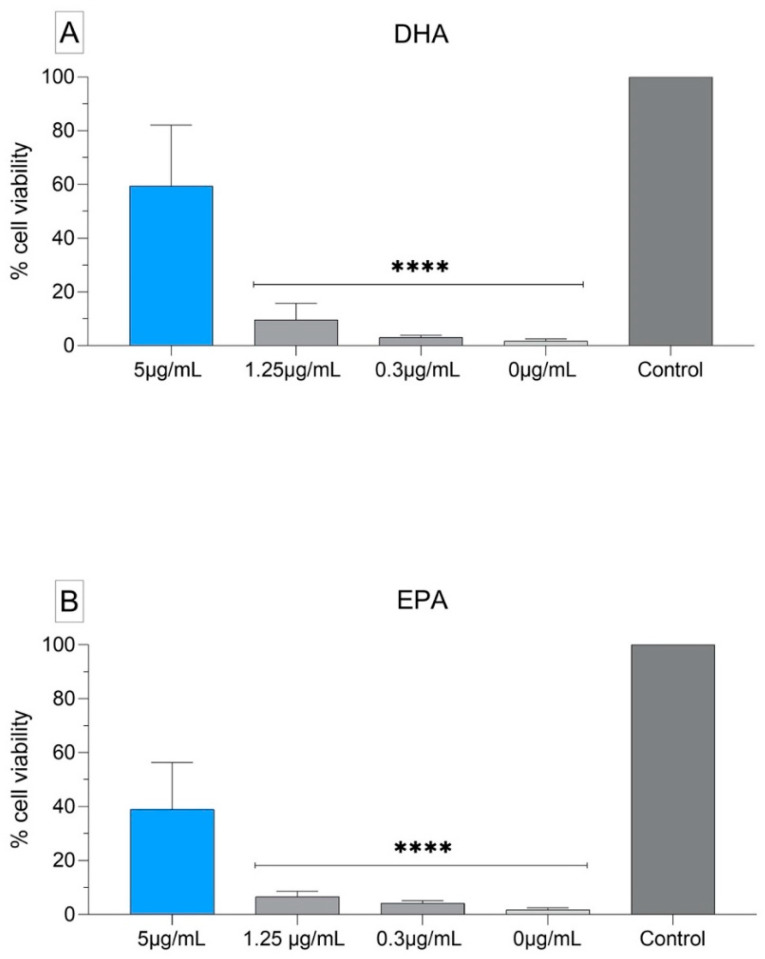
Cytotoxicity assay of EPA and DHA using human gingival fibroblasts HGF-1. The data are stated as mean ± SEM. (**A**) EPA; (**B**) DHA. The data obtained from the viability tests were analyzed using ordinary one-way ANOVA and Turkey’s multiple comparisons test (***** p* < 0.0001). The tests were made in duplicate.

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
