# Peer review of "Antibacterial and Anti-Biofilm Activity of Omega-3 Polyunsaturated Fatty Acids against Periprosthetic Joint Infections-Isolated Multi-Drug Resistant Strains"

_biomedicines, 2021, doi:10.3390/biomedicines9040334_

Round 1

Reviewer 1 Report

The paper details studies on 2 polyunsaturated fatty acids (PUFAs) and their inhibitory effect (bactericidal activity) on planktonic cells  and particularly biofilms/.  Laboratory strains of Staphylococcus aureus and S.epidermidis were tested as well as and coagulase negative species isolated from prosthetic joint infection. PUFAs are known to have antibacterial activity against Gram positive bacterial pathogens (staphylococci) and indeed have been shown in animal models to have in vivo efficacy in combination with antibiotics. The novel contribution of this paper is to demonstrate antibiofilm activity, both in reducing  the formation of biofilms in the presence of PUFAs in the growth medium, and the ability of PUFAs to kill bacteria in established biofilms.  This is an in vitro study which needs to be supported by investigating antibacterial activity in a suitable model of prosthetic joint infection before PUFAs can be put forward as biomedicines

Table 1 is unnecessary and should be relegated to supplemental information. What does a dash (-) mean ?

It would be helpful to state the number of CFUs inoculated into the broth containing the PUFAs (Figure 1).  It is not clear why the control (no PUFA) appears 5 times (with mg/L) when nothing is present. Is this a careless mistake?

There is no discussion as to why a concentration of 5 mg/L  EPA can support biofilm formation after incubation of bacteria overnight in the presence of the PUFA (Figure 3) while in Figure 1 the same concentration sterilizes the planktonic environment. Can bacterial cells avoid killing by forming biofilm?  Are there any bacteria surviving  on the surface of the 96 well plate in the form of  biofilms in Figure 1,

Minor issues

The paper should be carefully edited to eliminate grammatical mistakes, several of which occur in the abstract alone (line 13, 20,  27, 29).

Acronyms should be defined properly, particularly PJI which appears both in the title and abstract. There are two different acronyms for coagulase negative staphylococci in the abstract (CoNS and CNS)

Author Response

RESPONSE TO DECISION LETTER

Dear Prof. Dr. Jochen G. Hofstätter

Guest Editor: Journal – Biomedicines

Please find enclosed a carefully revised version of our manuscript entitled, “ANTIBACTERIAL AND ANTI-BIOFILM ACTIVITY OF OMEGA-3 POLYUNSATURATED FATTY ACIDS AGAINST PJI ISOLATED MULTI-DRUG RESISTANT STRAINS” - biomedicines-1149588” for re-submission to “Biomedicines”.

We were very pleased with the positive evaluation of our manuscript by the editors and the reviewers. However, the added some comments and suggestions which were all addressed in the revised version of our paper as detailed in the point to point response.

We hope that our paper is now acceptable for publication in your journal.

Yours sincerely,

Débora Cristina Coraça-Huber

Manuscript ID - biomedicines-1149588

Title - ANTIBACTERIAL AND ANTI-BIOFILM ACTIVITY OF OMEGA-3 POLYUNSATURATED FATTY ACIDS AGAINST PJI ISOLATED MULTI-DRUG RESISTANT STRAINS

Comments and Suggestions for Authors – REVIEW 1

The paper details studies on 2 polyunsaturated fatty acids (PUFAs) and their inhibitory effect (bactericidal activity) on planktonic cells  and particularly biofilms/.  Laboratory strains of Staphylococcus aureus and S.epidermidis were tested as well as and coagulase negative species isolated from prosthetic joint infection. PUFAs are known to have antibacterial activity against Gram positive bacterial pathogens (staphylococci) and indeed have been shown in animal models to have in vivo efficacy in combination with antibiotics. The novel contribution of this paper is to demonstrate antibiofilm activity, both in reducing  the formation of biofilms in the presence of PUFAs in the growth medium, and the ability of PUFAs to kill bacteria in established biofilms.  This is an in vitro study which needs to be supported by investigating antibacterial activity in a suitable model of prosthetic joint infection before PUFAs can be put forward as biomedicines

  • Thank you for the comments.

Table 1 is unnecessary and should be relegated to supplemental information. What does a dash (-) mean ?

  • The information about the dash was added and we are transferring the table to the supplemental data section.

It would be helpful to state the number of CFUs inoculated into the broth containing the PUFAs (Figure 1).  It is not clear why the control (no PUFA) appears 5 times (with mg/L) when nothing is present. Is this a careless mistake?

  • The information about the initial CFU concentration was added in the text and under the Figure 1 as suggested. The 5 times appearance of the control groups are because each control was treated differently with each concentration used also in other groups.  

There is no discussion as to why a concentration of 5 mg/L  EPA can support biofilm formation after incubation of bacteria overnight in the presence of the PUFA (Figure 3) while in Figure 1 the same concentration sterilizes the planktonic environment. Can bacterial cells avoid killing by forming biofilm?  Are there any bacteria surviving  on the surface of the 96 well plate in the form of  biofilms in Figure 1,

  • We improved the discussion section as suggested. The following discussion was added: Differently from the results with planktonic bacteria treated with EPA 5mg/L where complete kill was observed, some strains showed only reduction on biofilm formation when treated with this concentration. We associate this phenomenon to the fact that some strains produce more slime-rich biofilms, therefore, having more protection against antimicrobial substances. While PUFAs kill the bacterial cells by altering the fatty acids of the membrane, it may not influence the protein-rich matrix. This is an indication that higher doses of PUFAs may be tested for efficient anti-biofilm-activity or even the association of PUFAs with proteinases to increase the permeability in biofilms. 

Minor issues

The paper should be carefully edited to eliminate grammatical mistakes, several of which occur in the abstract alone (line 13, 20,  27, 29).

Acronyms should be defined properly, particularly PJI which appears both in the title and abstract. There are two different acronyms for coagulase negative staphylococci in the abstract (CoNS and CNS)

  • The entire manuscript was revised and the indicated mistakes corrected.

Reviewer 2 Report

In this manuscript the authors describe the application of DHA and EPA to be used as effective antimicrobial agents. They tested out the antibacterial activity of these compounds against different bacterial strains typically found in implant-associated infections. The antibacterial activity was performed in both planktonic and biofilm stage bacteria. The authors also determined the potential cytotoxicity of DHA and EPA on human gingival fibroblasts via MTT assay. Overall, all the experiments have been conducted logically and the results support the hypothesis. However, in its current stage, the manuscript cannot be accepted. I would recommend a major revision. Please make sure that the authors provide exact CFU reduction values in all of the results sections appropriately. They have provided in some sections but missed out in few sections. Below are my detailed comments:

  • Line 20 - it should be CoNS instead of CNS
  • line 28 - 'similar pathways were observed with clinical isolates' - please clarify this sentence
  • line 31 - 'as reference strains' - please correct this typo
  • line 32 - it should be CoNS instead of CNS
  • line 35 - ' EPA and DHA showed a certain protective effect' - this sentence does not provide accurate result analysis. Please rephrase
  • section 2.2 - please italicize all the bacterial names. Also, provide the total number of reference isolates and clinical isolates used in this study
  • line 142 - do the authors mean 2,000 x g?
  • line 146/147 - which technique did the authors use for quantification? Spread-plate or drop-plate?
  • Line 162 - mention the frequency of the sonicator
  • Line 163 - which technique did the authors use for quantification? Spread-plate or drop-plate?
  • Line 179 - which technique did the authors use for quantification? Spread-plate or drop-plate?
  • Section 2.8 - how is the data represented throughout the manuscript? Is it mean ± SD or mean ± SEM?
  • Section 3.2 - provide actual CFU reduction numbers wherever you can
  • Figure 1 - Please denote (*) to the data sets where significant results were achieved. Please write a sentence in figure legend as to how is the data represented. Is it mean ± SD or mean ± SEM? What is the n-value? Fig.1b - italicize bacteria name
  • Figure 2 - Please denote (*) to the data sets where significant results were achieved. Please write a sentence in figure legend as to how is the data represented. Is it mean ± SD or mean ± SEM? What is the n-value?
  • Figure 3 - Please denote (*) to the data sets where significant results were achieved. Please write a sentence in figure legend as to how is the data represented. Is it mean ± SD or mean ± SEM? What is the n-value? Also, it is advisable to change the y-axis to CFU/cm2 since biofilm was formed on discs.
  • Figure 4 - Please denote (*) to the data sets where significant results were achieved. Please write a sentence in figure legend as to how is the data represented. Is it mean ± SD or mean ± SEM? What is the n-value? Also, it is advisable to change the y-axis to CFU/cm2 since biofilm was formed on discs.
  • Figure 5 - Please denote (*) to the data sets where significant results were achieved. Please write a sentence in figure legend as to how is the data represented. Is it mean ± SD or mean ± SEM? What is the n-value? Also, it is advisable to change the y-axis to CFU/cm2 since biofilm was formed on discs.
  • Line 420 - 'a light decrease of viability' - please provide actual numbers for viability data.
  • Figure 6 - Presenting the viability data based on OD values is not a proper way for representing the cell-viability data. I would recommend the authors to convert the y-axis to % cell-viability so that it becomes easier for the readers to understand the graph.

Author Response

RESPONSE TO DECISION LETTER

Dear Prof. Dr. Jochen G. Hofstätter

Guest Editor: Journal – Biomedicines

Please find enclosed a carefully revised version of our manuscript entitled, “ANTIBACTERIAL AND ANTI-BIOFILM ACTIVITY OF OMEGA-3 POLYUNSATURATED FATTY ACIDS AGAINST PJI ISOLATED MULTI-DRUG RESISTANT STRAINS” - biomedicines-1149588” for re-submission to “Biomedicines”.

We were very pleased with the positive evaluation of our manuscript by the editors and the reviewers. However, the added some comments and suggestions which were all addressed in the revised version of our paper as detailed in the point to point response.

We hope that our paper is now acceptable for publication in your journal.

Yours sincerely,

Débora Cristina Coraça-Huber

Manuscript ID - biomedicines-1149588

Title - ANTIBACTERIAL AND ANTI-BIOFILM ACTIVITY OF OMEGA-3 POLYUNSATURATED FATTY ACIDS AGAINST PJI ISOLATED MULTI-DRUG RESISTANT STRAINS

Comments and Suggestions for Authors – REVIEW 2

In this manuscript the authors describe the application of DHA and EPA to be used as effective antimicrobial agents. They tested out the antibacterial activity of these compounds against different bacterial strains typically found in implant-associated infections. The antibacterial activity was performed in both planktonic and biofilm stage bacteria. The authors also determined the potential cytotoxicity of DHA and EPA on human gingival fibroblasts via MTT assay. Overall, all the experiments have been conducted logically and the results support the hypothesis. However, in its current stage, the manuscript cannot be accepted. I would recommend a major revision. Please make sure that the authors provide exact CFU reduction values in all of the results sections appropriately. They have provided in some sections but missed out in few sections. Below are my detailed comments:

  • Thank you for your comments.
  • Line 20 - it should be CoNS instead of CNS
  • The information was corrected in the manuscript.
  • line 28 - 'similar pathways were observed with clinical isolates' - please clarify this sentence
  • The information was corrected in the manuscript.
  • line 31 - 'as reference strains' - please correct this typo
  • The information was inadequate and therefore deleted from the abstract.
  • line 32 - it should be CoNS instead of CNS
  • The information was corrected in the manuscript.
  • line 35 - ' EPA and DHA showed a certain protective effect' - this sentence does not provide accurate result analysis. Please rephrase
  • The information was corrected in the manuscript.
  • section 2.2 - please italicize all the bacterial names. Also, provide the total number of reference isolates and clinical isolates used in this study
  • The information was corrected in the manuscript.
  • line 142 - do the authors mean 2,000 x g?
  • Yes, the information was corrected in the manuscript.
  • line 146/147 - which technique did the authors use for quantification? Spread-plate or drop-plate?
  • The information was added in the manuscript.
  • Line 162 - mention the frequency of the sonicator
    • The information was added in the manuscript.
  • Line 163 - which technique did the authors use for quantification? Spread-plate or drop-plate?
  • The information was added in the manuscript.
  • Line 179 - which technique did the authors use for quantification? Spread-plate or drop-plate?
  • The information was added in the manuscript.
  • Section 2.8 - how is the data represented throughout the manuscript? Is it mean ± SD or mean ± SEM?
  • The information was added in the manuscript.

  • Section 3.2 - provide actual CFU reduction numbers wherever you can
  • The information about the initial CFU concentration used was added in the manuscript.
  • Figure 1 - Please denote (*) to the data sets where significant results were achieved. Please write a sentence in figure legend as to how is the data represented. Is it mean ± SD or mean ± SEM? What is the n-value? Fig.1b - italicize bacteria name
  • Figure 2 - Please denote (*) to the data sets where significant results were achieved. Please write a sentence in figure legend as to how is the data represented. Is it mean ± SD or mean ± SEM? What is the n-value?
  • Figure 3 - Please denote (*) to the data sets where significant results were achieved. Please write a sentence in figure legend as to how is the data represented. Is it mean ± SD or mean ± SEM? What is the n-value? Also, it is advisable to change the y-axis to CFU/cm2 since biofilm was formed on discs.
  • Figure 4 - Please denote (*) to the data sets where significant results were achieved. Please write a sentence in figure legend as to how is the data represented. Is it mean ± SD or mean ± SEM? What is the n-value? Also, it is advisable to change the y-axis to CFU/cm2 since biofilm was formed on discs.
  • Figure 5 - Please denote (*) to the data sets where significant results were achieved. Please write a sentence in figure legend as to how is the data represented. Is it mean ± SD or mean ± SEM? What is the n-value? Also, it is advisable to change the y-axis to CFU/cm2 since biofilm was formed on discs.
  • The information was corrected in the manuscript.
  • Line 420 - 'a light decrease of viability' - please provide actual numbers for viability data.
  • The information was corrected in the manuscript.
  • Figure 6 - Presenting the viability data based on OD values is not a proper way for representing the cell-viability data. I would recommend the authors to convert the y-axis to % cell-viability so that it becomes easier for the readers to understand the graph.
  • The information was corrected in the manuscript.

Round 2

Reviewer 1 Report

The authors have revised the manuscript satisfactorily, apart from retaining the acronym PJI in the title

Author Response

RESPONSE TO DECISION LETTER

Dear Prof. Dr. Jochen G. Hofstätter

Guest Editor: Journal – Biomedicines

Please find enclosed a carefully revised version of our manuscript entitled, “ANTIBACTERIAL AND ANTI-BIOFILM ACTIVITY OF OMEGA-3 POLYUNSATURATED FATTY ACIDS AGAINST PJI ISOLATED MULTI-DRUG RESISTANT STRAINS” - biomedicines-1149588” for re-submission to “Biomedicines”.

We were very pleased with the positive evaluation of our manuscript by the editors and the reviewers. However, the added some comments and suggestions which were all addressed in the revised version of our paper as detailed in the point to point response.

We hope that our paper is now acceptable for publication in your journal.

Yours sincerely,

Débora Cristina Coraça-Huber

Manuscript ID - biomedicines-1149588

Title - ANTIBACTERIAL AND ANTI-BIOFILM ACTIVITY OF OMEGA-3 POLYUNSATURATED FATTY ACIDS AGAINST PJI ISOLATED MULTI-DRUG RESISTANT STRAINS

Comments and Suggestions for Authors – REVIEW 1

The authors have revised the manuscript satisfactorily, apart from retaining the acronym PJI in the title

- We apologise for this misunderstanding and corrected the title as suggested by the reviewer 1. 

Reviewer 2 Report

The authors have still not completely responded with to my comments. 

1) Section 3.2 - you need to provide actual CFU reduction numbers instead of writing "reduced cell-count"

2) Figures 3, 4, and 5 - it is advisable to change the y-axis to CFU/cm2 since biofilm was formed on discs

3) Figure 6 - Presenting the viability data based on OD values is not a proper way for representing the cell-viability data. I would recommend the authors to convert the y-axis to % cell-viability so that it becomes easier for the readers to understand the graph.

Author Response

RESPONSE TO DECISION LETTER

Dear Prof. Dr. Jochen G. Hofstätter

Guest Editor: Journal – Biomedicines

Please find enclosed a carefully revised version of our manuscript entitled, “ANTIBACTERIAL AND ANTI-BIOFILM ACTIVITY OF OMEGA-3 POLYUNSATURATED FATTY ACIDS AGAINST PJI ISOLATED MULTI-DRUG RESISTANT STRAINS” - biomedicines-1149588” for re-submission to “Biomedicines”.

We were very pleased with the positive evaluation of our manuscript by the editors and the reviewers. However, the added some comments and suggestions which were all addressed in the revised version of our paper as detailed in the point to point response.

We hope that our paper is now acceptable for publication in your journal.

Yours sincerely,

Débora Cristina Coraça-Huber

Manuscript ID - biomedicines-1149588-4

Title - ANTIBACTERIAL AND ANTI-BIOFILM ACTIVITY OF OMEGA-3 POLYUNSATURATED FATTY ACIDS AGAINST PJI ISOLATED MULTI-DRUG RESISTANT STRAINS

Comments and Suggestions for Authors – REVIEW 2

The authors have still not completely responded with to my comments. 

- we apologise and are submitting the responses for each comment below:

1) Section 3.2 - you need to provide actual CFU reduction numbers instead of writing "reduced cell-count"

- the CFU reduction values were inserted in section 3.2 as requested. 

2) Figures 3, 4, and 5 - it is advisable to change the y-axis to CFU/cm2 since biofilm was formed on discs

- The figure y-axis were corrected as sugested 

3) Figure 6 - Presenting the viability data based on OD values is not a proper way for representing the cell-viability data. I would recommend the authors to convert the y-axis to % cell-viability so that it becomes easier for the readers to understand the graph.

- The graph was corrected